# Transient protein-protein interactions perturb *E. coli* metabolome and cause gene dosage toxicity

Sanchari Bhattacharyya[1†], Shimon Bershtein[2†], Jin Yan[1,3], Tijda Argun[1], Amy I Gilson[1], Sunia A Trauger[4], Eugene I Shakhnovich[1*]

[1]Department of Chemistry and Chemical Biology, Harvard University, Cambridge, United States; [2]Department of Life Sciences, Ben-Gurion University of the Negev, Beer-Sheva, Israel; [3]College of Chemical Engineering, Sichuan University, Chengdu, China; [4]Small Molecule Mass Spectrometry, Northwest Laboratories, Harvard University, Cambridge, United States

*For correspondence:
shakhnovich@chemistry.harvard.edu

[†]These authors contributed equally to this work

**Competing interests:** The authors declare that no competing interests exist.

**Abstract** Gene dosage toxicity (GDT) is an important factor that determines optimal levels of protein abundances, yet its molecular underpinnings remain unknown. Here, we demonstrate that overexpression of DHFR in *E. coli* causes a toxic metabolic imbalance triggered by interactions with several functionally related enzymes. Though deleterious in the overexpression regime, surprisingly, these interactions are beneficial at physiological concentrations, implying their functional significance *in vivo*. Moreover, we found that overexpression of orthologous DHFR proteins had minimal effect on all levels of cellular organization – molecular, systems, and phenotypic, in sharp contrast to *E. coli* DHFR. Dramatic difference of GDT between '*E. coli*'s self' and 'foreign' proteins suggests the crucial role of evolutionary selection in shaping protein-protein interaction (PPI) networks at the whole proteome level. This study shows how protein overexpression perturbs a dynamic metabolon of weak yet potentially functional PPI, with consequences for the metabolic state of cells and their fitness.

## Introduction

Experimental approaches to mapping the functional relationships between genes and phenotypes traditionally use perturbations of gene dosage via systematic deletion (*Baba et al., 2006*; *Giaever et al., 2002*; *Pan et al., 2004*; *Sopko et al., 2006*), down-regulation (*Mnaimneh et al., 2004*), or overexpression (*Sopko et al., 2006*; *Gelperin et al., 2005*; *Kitagawa et al., 2005*; *Makanae et al., 2013*) of the target genes. Phenotypes produced by gene overexpression generate a plethora of fitness effects that tend to deviate from those observed in gene deletion studies (*Sopko et al., 2006*; *Prelich, 2012*). High-throughput studies revealed that overexpression of a substantial fraction of genes is detrimental to fitness. In *E. coli*, overexpression of the majority of proteins is mildly to severely toxic under the conditions of the experiment (*Kitagawa et al., 2005*), whereas in yeast around 15% of overexpressed proteins produce morphological changes and drop in growth (*Sopko et al., 2006*). These findings are intriguing, given that gene-dosage increase plays a central role in evolutionary adaptations (*Kondrashov, 2012*), such as adaptations to a different carbon source (*Brown et al., 1998*), temperature (*Riehle et al., 2001*), and emergence of antibiotic resistance (*Andersson and Hughes, 2009*).

Several mechanisms were proposed to explain gene dosage toxicity (GDT), including resource overload (*Makanae et al., 2013*; *Shachrai et al., 2010*; *Snoep et al., 1995*; *Stoebel et al., 2008*), aggregation toxicity (*Geiler-Samerotte et al., 2011*; *Kaiser et al., 2013*), stoichiometric imbalance

(*Papp et al., 2003*; *Veitia et al., 2008*), and non-specific PPIs (*Ma et al., 2010*; *Vavouri et al., 2009*). Vavouri *et al* hypothesized that GDT in yeast is predominantly caused by disordered proteins because of their potential involvement in multiple protein-protein and protein-DNA interactions, which, under the overexpression regime, could bring about deleterious mis-interactions (*Vavouri et al., 2009*). Singh and Dash hypothesized that electrostatic mis-interactions might be responsible for GDT in bacteria, which, unlike yeast, mostly lack proteins with disordered regions (*Singh and Dash, 2013*). Theoretical analyses suggested that the balance between functional and non-functional protein-protein interactions (PPI) is an important determinant of protein abundances in the cell (*Deeds et al., 2007*; *Zhang et al., 2008*; *Heo et al., 2011*; *Wallace and Drummond, 2015*). Thus, it is plausible that GDT might be caused by protein mis-interactions. However, specific mechanisms by which overexpression-induced protein mis-interactions cause toxicity and loss of fitness are not known.

Here we elucidate a molecular mechanism of GDT by focusing on molecular, systems, and organismal effects of overexpression of *E. coli* dihydrofolate reductase (EcDHFR). DHFR is an essential enzyme that catalyzes electron transfer reaction to form tetrahydrofolate, a carrier of single-carbon functional groups utilized in specific biochemical reactions (*Harvey and Dev, 1975*; *Schnell et al., 2004*). We explored fitness of *E. coli* in a broad range of EcDHFR abundances – from strong down-regulation to ~850 fold overexpression and established that the basal expression level is close to the optimal at which fitness is the highest, while *both* down regulation and overexpression appear toxic. While the drop in *E. coli* fitness upon DHFR downregulation is predicted by the enzymatic flux kinetics analysis (*Bershtein et al., 2015*; *Rodrigues, 2016*), toxicity upon overexpression is at variance with enzymatic flux kinetics analysis, which predicts fitness neutrality once the functional capacity of an enzyme, defined as the product of its intracellular abundance and catalytic efficiency ($k_{cat}/K_M$), exceeds the threshold flux through a metabolic path (*Feist and Palsson, 2010*; *Lewis et al., 2012*; *Kacser and Burns, 1981*; *Dykhuizen et al., 1987*).

In order to get a detailed insight into the mechanism of GDT, we systematically analyzed perturbations in protein-protein interaction (PPI) and metabolic networks caused by overexpression of EcDHFR and its bacterial orthologues. We found that metabolic and fitness effects of overexpression were triggered exclusively by EcDHFR, while the effects of overexpressing DHFR orthologues were much weaker on all scales. Pull-down assay coupled to LC-MS/MS revealed that over-expressed DHFR from *E. coli* and other bacteria interact with several enzymes in DHFR functional vicinity (folate pathway/purine biosynthesis), and *in vitro* analysis confirmed the presence of these weak interactions. However, the mutual effects on activity of DHFR and its interacting partners were markedly different between EcDHFR and its orthologues. These findings illustrate functional significance and ensuing evolutionary selection of weak transient protein-protein interactions in the crowded cytoplasm.

## Results

### *E. coli* is highly sensitive to DHFR dosage variation

Tetrahydrofolate, the product of DHFR activity, is utilized in numerous one-carbon metabolism reactions, including de novo purine biosynthesis, dTTP formation, and methionine and glycine production (*Harvey and Dev, 1975*; *Schnell et al., 2004*). As expected, a drop in DHFR activity and/or abundance, induced by either genetic changes (*Bershtein et al., 2015*, *2013*, *2012*), or DHFR inhibition (*Rodrigues, 2016*; *Kwon et al., 2008*; *Sangurdekar et al., 2011*), results in reduced fitness. But what is the effect of an *increase* in DHFR dosage on fitness? To answer this question, we designed an experimental system capable of generating controlled variation in a broad range of intracellular DHFR abundance in *E. coli*, and measured the effect of these changes on growth (*Figure 1A*). A controlled drop in DHFR production was achieved by introducing LacI binding sites (*lacO*) in the chromosomal regulatory region of *folA* gene in a genetically modified *E. coli* strain constitutively expressing LacI (see Materials and methods and [*Bershtein et al., 2013*]). Dialing down IPTG concentration (LacI repressor) resulted in a gradual decrease (over 100 fold) of the intracellular DHFR abundance (*Figure 1A* and *Figure 1—source data 1*). A gradual increase in DHFR abundance (from ~5 to 850-fold) was achieved by transforming WT *E. coli* strain with a plasmid carrying *folA* gene under the control of arabinose inducible pBAD promoter (see Materials and methods and

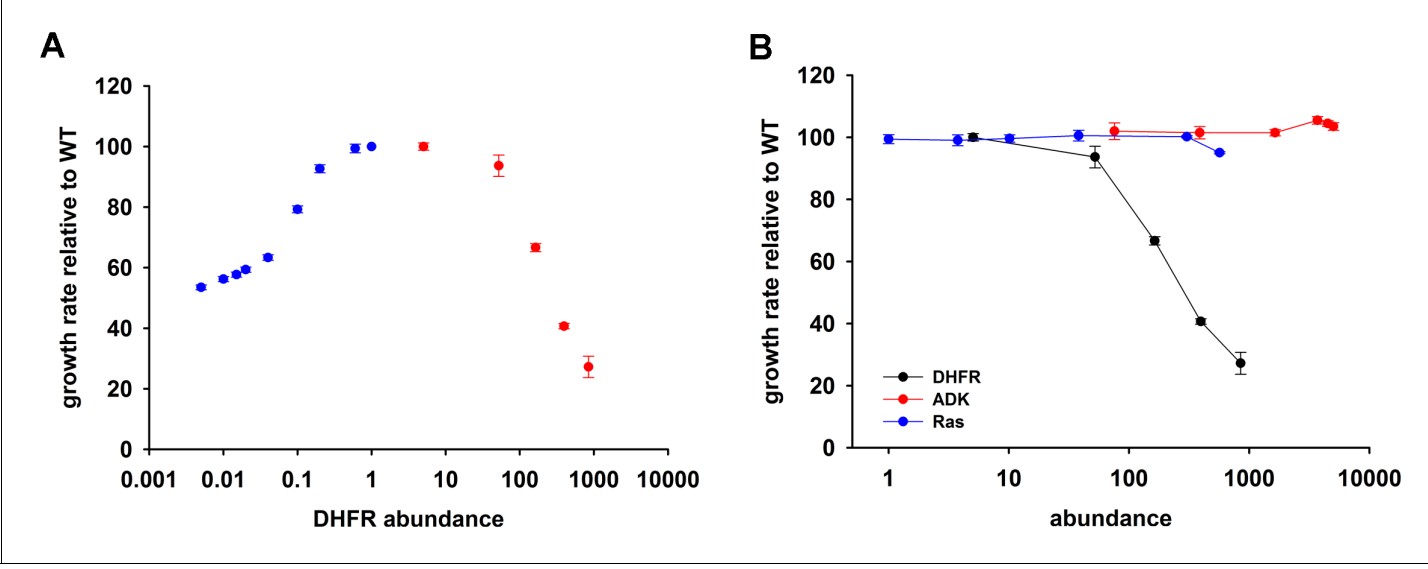

**Figure 1.** Over-expression of endogenous DHFR is detrimental to bacterial growth. (A) The effect of variation in DHFR dosage on fitness of *E. coli*. Change in the intracellular DHFR abundance (in log scale) is plotted against growth rate. DHFR abundance and bacterial growth are normalized against the parameters observed for wild-type strain. In case of over-expression, the growth rates were normalized with values obtained with no inducer (arabinose) (also see *Figure 1—source data 1*). Controlled drop (over 100 fold) in DHFR abundance (blue circles) was induced by IPTG titration of the chromosomal *folA* gene of *E. coli* MG1655 strain at 30°C which was modified to contain LacI binding sites (see Materials and methods and [*Bershtein et al., 2013*]) and shows a Michaelis-Menten type dependence on fitness described in (*Bershtein et al., 2013*). A controlled increase (~850 fold) in DHFR abundance (red circles) was achieved by arabinose titration of *E. coli* BW27783 strain transformed with a plasmid at 37°C carrying the endogenous *folA* gene under the control pBAD promoter. DHFR over-expression shows a strong dose-dependent drop in fitness (Spearman r = −1, p=0.0167). The obtained abundance vs fitness function shows that the basal endogenous DHFR levels approach a physiological optimum with respect to *E. coli* growth rate. Overexpression of C-terminal His-tagged EcDHFR generated identical fitness drop (*Figure 4—figure supplement 2*). (B) Fitness as a function of protein abundance shown for EcDHFR, *E. coli* Adenylate Kinase (ADK) and a eukaryotic non-endogenous protein H-ras p21 (Ras). Only the expression of DHFR shows a dose-dependent toxicity and, therefore, it is not a generic effect of an over-expression burden. The intracellular DHFR, Ras, and ADK abundances were measured by Western Blot with custom raised antibodies (also see Materials and methods).

The following source data and figure supplement are available for figure 1:

**Source data 1.** Growth rate and abundance for over-expression (37°C) and down-regulation(30°C) of DHFR.

**Figure supplement 1.** Growth rate (relative to untransformed cells) of *E.coli* cells transformed with an empty pBAD plasmid over a range of arabinose concentrations.

*Figure 1A,B* and *Figure 1—source data 1*). *Figure 1A* shows that both drop *and* increase in intracellular DHFR abundance result in severe reduction (up to 73%) in growth rate. Fitness as a function of DHFR abundance therefore shows an 'optimum' at the basal DHFR level. However, the shape of the DHFR abundance – fitness function appears to be very different between DHFR depletion and DHFR overexpression regimes (*Figure 1A*). A drop in fitness upon DHFR depletion is well described by Michaelis-Menten like dependence predicted by metabolic flux kinetics (*Kacser and Burns, 1981*; *Dykhuizen et al., 1987*) and verified experimentally (*Bershtein et al., 2015*; *Rodrigues, 2016*; *Bershtein et al., 2013*). In contrast, the increase in the intracellular DHFR levels above the optimal (basal) level leads to a sigmoidal decline in fitness across the entire measured range of DHFR abundances (*Figure 1A,B*). The qualitative differences between DHFR depletion and overexpression regimes clearly indicate that different mechanisms are responsible for loss of fitness at higher and lower ends of DHFR abundance-fitness curve.

## DHFR dosage toxicity is not a result of resource overload or aggregation toxicity

Resource overload, or 'protein burden', is often invoked to explain the dosage toxicity of endogenous enzymes (*Moriya, 2015*), e.g. toxicity from lacZ production in the absence of lactose in *E. coli* (*Dekel and Alon, 2005*), and overexpression toxicity of glycolytic enzymes in yeast (*Makanae et al., 2013*) and *Z. mobilis* (*Stoebel et al., 2008*). To address the possibility that the observed toxicity of DHFR overexpression is a result of protein expression burden, we measured the effect of expression of two other control proteins on bacterial growth under identical conditions: an endogenous essential protein adenylate kinase (ADK), and a non-endogenous human H-ras p21 (Ras), - a small globular protein with size and fold similar to DHFR (*de Vos et al., 1988*) (*Figure 1B*) At highest inducer concentration, Ras reached the abundance that is ~600 fold greater than that of the endogenous DHFR, while ADK copy number reached ~5000 fold increase over the physiological DHFR levels without causing any substantial drop in growth. Further, no toxicity was observed from an empty pBAD plasmid (*Figure 1—figure supplement 1*). Since *E. coli* DHFR and the control proteins are produced to comparable levels and from the same pBAD expression system, we conclude that the observed GDT of DHFR is protein specific and cannot be explained by either energetic or competition components of resource overload.

Fitness cost of overexpression is often attributed to aggregation of an overexpressed protein (*Geiler-Samerotte et al., 2011*). To test for possible aggregation of DHFR *in vivo*, *folA* gene in pBAD plasmid was fused in frame with the GFP coding gene and cellular fluorescence was analyzed. All overexpressing cells presented a highly diffuse fluorescence pattern indicative of a lack of aggregation (*Figure 2A*). A Western Blot analysis was carried out with the *E. coli* cell lysate containing pBAD-DHFR-GFP to confirm that the observed fluorescence emanated from the DHFR-GFP fusion protein and not from GFP alone due to a possible proteolysis around the linker region of the fusion protein (*Figure 2B,C*). Thus, aggregation toxicity cannot cause GDT of DHFR overexpression.

## Overexpression of *E. coli* DHFR triggers a metabolic imbalance, while overexpression of DHFR bacterial orthologues does not

A subset of metabolic enzymes involved in more than one reaction can cause a bottleneck effect in some of the metabolic branches upon overexpression (*Wagner et al., 2013*). Although DHFR is a single-reaction enzyme, we tested whether its overexpression toxicity can be explained by a metabolic cost. To this end, we performed a metabolomics analysis of *E. coli* cultures overexpressing endogenous DHFR and ADK as a negative control. Whereas no measurable perturbation in metabolite levels could be detected upon ADK overexpression, a strong shift in metabolite pools was observed with overexpressed DHFR (*Figure 3A* and *Figure 4—source data 1* (Sheet 1 and 2)). Specifically, we found a pronounced increase in AICAR and dUMP levels, accompanied by a drop in the levels of purines and pyrimidines (dTMP, dTTP, AMP, GMP, IMP). Toxicity could be partially rescued by adding these depleted metabolites (IMP, dTMP) to the growth medium (*Figure 3B* and *Figure 3—figure supplement 1*).

Next, we tested the effect of overexpression of six distantly related but highly active DHFR orthologues from several bacterial species (*Bershtein et al., 2015*, *2013*) (*Figure 4—figure supplement 1* and *Figure 4—source data 2*). Remarkably, we found that these DHFRs had only a marginal effect on *E. coli* fitness (*Figure 4A,B*). Furthermore, we found that over-expression of DHFRs originating from *Listeria innocua* (DHFR6) and *Bordetella avium* (DHFR11) did not result in massive changes in metabolite levels as found for *E. coli* DHFR (*Figure 4C* and *Figure 4—source data 1* (Sheet 1 and 2)). A small drop in IMP levels was observed for both, however AICAR levels were unperturbed (*Figure 4C*). Moreover, growth rate of all strains were highly correlated with their intracellular dTTP levels, indicating that depletion of metabolite levels is a direct signature of toxicity (*Figure 4D*). In summary, loss of fitness appears to be directly related to metabolite imbalance although the excessive DHFR activity per se is not the root cause of this imbalance.

## Interacting partners of endogenous and orthologous DHFRs are enriched in 1-carbon metabolism enzymes

An increase in the intracellular concentration of proteins can induce potentially detrimental non-specific 'mis-interactions' that are not present at the level of endogenous abundances (*Vavouri et al.,*

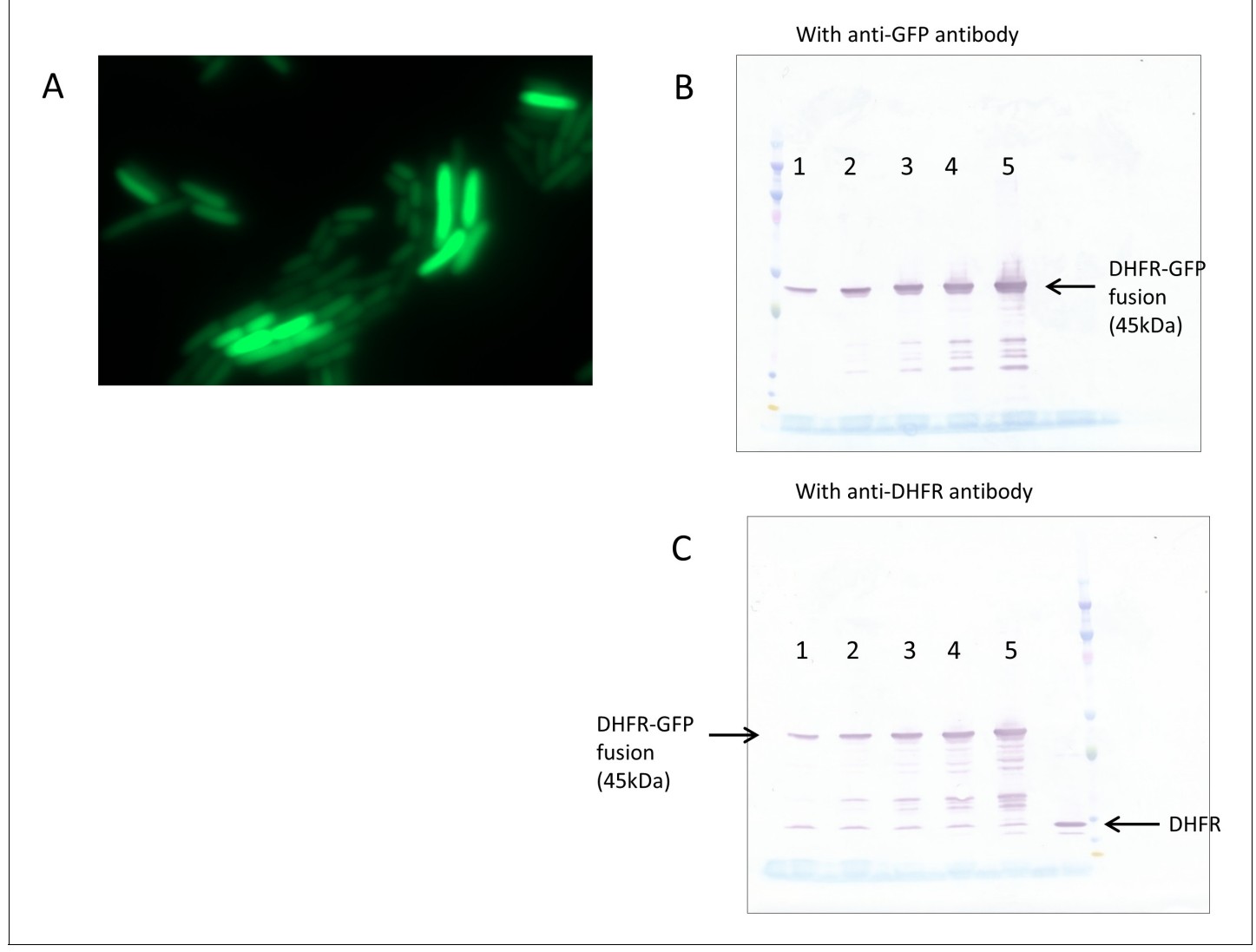

**Figure 2.** Over-expressed DHFR does not aggregate inside the cell. (**A**) DHFR fused to monomeric superfolder GFP was expressed from pBAD plasmid, and fluorescence of the cells was detected using a Zeiss Cell Observer microscope. Intracellular abundance of the fusion protein in the image shown was ~350 fold over chromosomally expressed DHFR. The fluorescence was uniformly spread over the cells, ruling out any substantial aggregation. Over-expressed fusion protein is not cleaved inside the cell, as both (**B**) anti-GFP and (**C**) anti-DHFR antibodies showed very minor fraction of cleaved proteins. In panels (**B**) and (**C**), lanes 1 to 5 represent different expression levels of DHFR-GFP fusion protein.

*2009*; *Zhang et al., 2008*; *Heo et al., 2011*; *Maslov and Ispolatov, 2007*). Importantly, *E. coli* DHFR is a low copy monomeric enzyme (*Taniguchi et al., 2010*) with no physical interactions detected with other proteins when used as a prey at its basal level (*Hu et al., 2009*). Further, no DHFR interactions were detected in yeast, and only one in Drosophila [dip.doe-mbi.ucla.edu]. We sought to determine possible interacting partners of over-expressed EcDHFR and two controls, ADK and Ras. To that end, protein complexes in cells were stabilized *in vivo* by a cell-permeable cross-linker, followed by cell lysis, immunoprecipitation (IP) performed on soluble lysates with polyclonal rabbit anti-EcDHFR, ADK, and Ras antibodies, and LC-MS/MS analysis (See *Figure 5A* and Materials and methods for details). As shown in *Figure 5B*, *Figure 5—source data 1* and *Figure 5—source data 2*, overexpressed DHFR pulled out several interaction partners, while ADK and Ras picked very few (3–4), suggesting that cellular PPI is the most likely reason for the DHFR overexpression toxicity. We also compared the PPI profile of *E. coli* DHFR with its orthologue from *Bordetella avium* (DHFR11). Since anti-EcDHFR antibodies do not bind DHFR 11 due to a large sequence

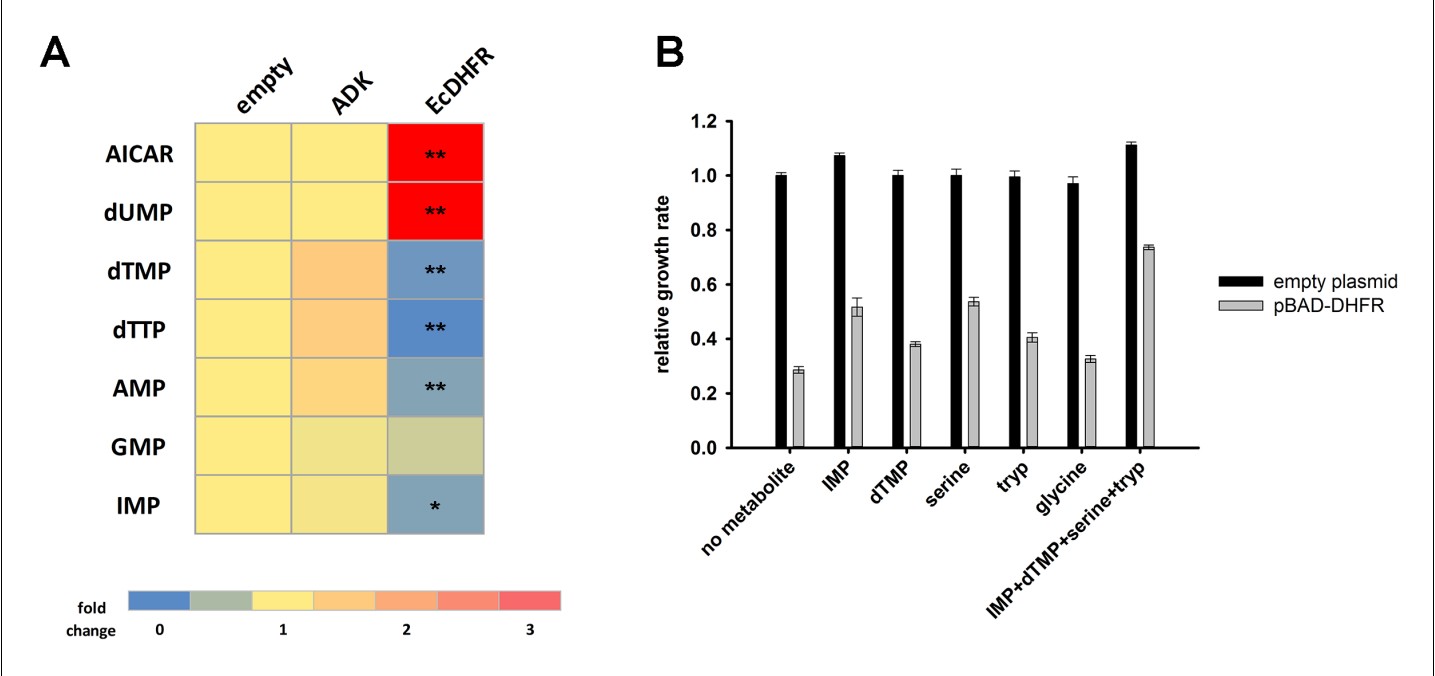

**Figure 3.** Overexpression of endogenous DHFR triggers a major metabolic shift. (**A**) Heat-map of the intracellular levels of various metabolites detected in bacterial cells over-expressing endogenous ADK and DHFR proteins from a pBAD plasmid. While ADK over-expression does not alter the metabolite levels, DHFR over-expression results in a pronounced up-regulation of AICAR and dUMP levels, and down-regulation of purines and pyrimidine nucleotides (dTMP, dTTP, AMP, IMP) (See Materials and methods). In all experiments, sample size was three biological replicates. For statistical significance, levels in DHFR over-expression were compared to ADK over-expression, where * denotes p-value<0.05 and ** denotes p-value<0.001. (**B**) Metabolic complementation by addition of purine and pyrimidine nucleotides along with several amino acids, like serine and tryptophan, results in a partial rescue of fitness. 1 mM of each of the metabolites was added to supplemented M9 medium at 37°C. Growth rates under different conditions were normalized by the growth rate obtained with empty plasmid without addition of any metabolite, and was referred to as 'relative growth rate'. For all experiments the arabinose concentration used was 0.05%, which corresponds to ~850 fold over-expression of DHFR and ~5000 fold increase of ADK.

The following figure supplement is available for figure 3:

**Figure supplement 1.** Overlay of growth curves of *E.coli* over-expressing DHFR and those in the presence of various metabolites.

distance, His-tagged versions of both proteins were used in conjunction with mouse monoclonal anti-His antibodies (see Materials and methods). As a first step, we compared the PPI profiles of EcDHFR obtained with two different antibodies (rabbit polyclonal anti-EcDHFR and mouse monoclonal anti-His antibodies), and found that the anti-His antibody pulled out a smaller number of interacting proteins (*Figure 5—source data 1*). This is probably because cellular PPI that target regions close to the N-terminal His-tag of DHFR cause steric hindrance for binding of anti-His antibodies. Despite this caveat, we went ahead to compare the PPI profiles of EcDHFR and orthologous DHFR 11 and found that DHFR 11 pulled out many more proteins than EcDHFR (*Figure 5—source data 1* and *Figure 5—source data 2*). Interestingly, almost all proteins constituting the EcDHFR interactome can also be found in the DHFR11 interactome. Why is then overexpression toxicity limited only to EcDHFR? A closer look into the set of 29 interacting proteins of EcDHFR that were common to both anti-EcDHFR and anti-His antibody pull-downs (Sheet 1 of *Figure 5—source data 2*) allowed us to classify them according to their function (Sheet 2 of *Figure 5—source data 2*). While several proteins belong to carbohydrate metabolism and amino acid biosynthesis, there were several enzymes closely related to DHFR function, such as 1-carbon metabolism. Tetrahydrofolate, the product of DHFR is immediately taken up by serine hydroxymethyltransferase (GlyA) to produce 5,10-methylene-THF, which in a series of steps is subsequently used to produce dTMP from dUMP (*Figure 6B*). 5,10-methylene-THF is also utilized in a second pathway to produce [10]Nf-THF, which is then

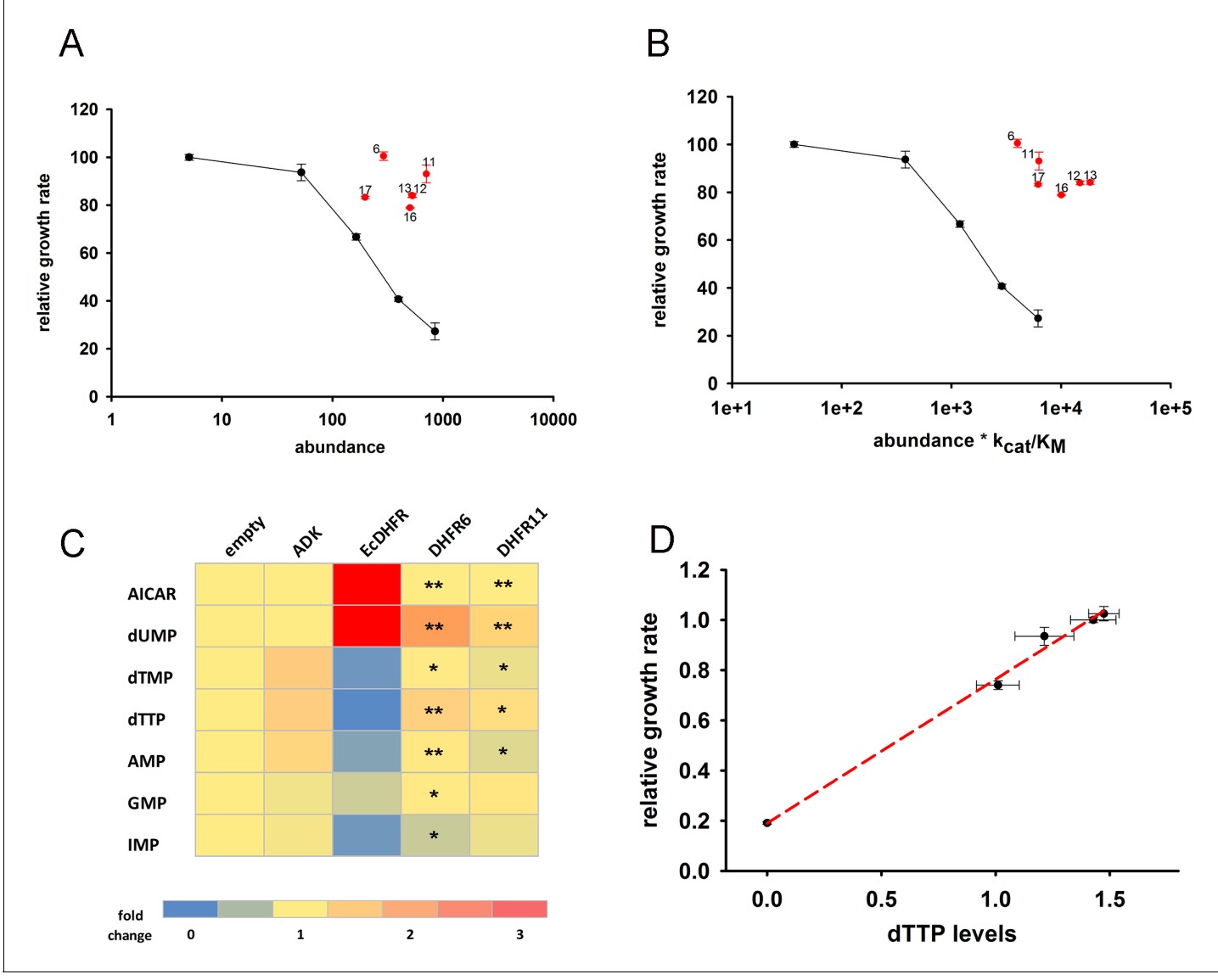

**Figure 4.** Orthologous DHFR proteins do not cause metabolic imbalance and dosage-related toxicity. Fitness is shown as a function of (**A**) intracellular DHFR abundance and (**B**) (abundance * $k_{cat}/K_M$) for overexpression of *E. coli* DHFR as well as those from five mesophilic bacteria *Listeria innocua* (DHFR 6), *Bordetella avium* (DHFR 11), *Leuconostoc mesenteroides* (DHFR 12), *Aeromonas hydrophila* (DHFR 13), *Clostridium cellulolyticum* (DHFR 16) and *Streptococcus dysgalactiae* (DHFR 17) expressed from the same plasmid under pBAD-promoter using an inducer concentration of 0.05%. Fitness of cells over-expressing orthologous DHFRs was significantly different from those expressing *E. coli* DHFR at equivalent concentrations (in all cases, p-value<0.001) (also see Materials and methods) and only endogenous EcDHFR was highly toxic to *E. coli*. The intracellular abundance of orthologous DHFR was measured by Western Blot with anti-His antibodies (See Materials and methods). Overexpression of orthologous DHFR proteins lacking the His-tag produced identical results (***Figure 4—figure supplement 2***). (**C**) Heat-map of the intracellular levels of various metabolites detected in cells overexpressing endogenous ADK and *E. coli* DHFR as well as two orthologous DHFRs (6 and 11). An empty pBAD plasmid is used as a control. Metabolite levels in orthologous DHFR over-expression were compared to EcDHFR over-expression for statistical significance, where * denotes p-value<0.05 and ** denotes p-value<0.001. Hence when compared to *E. coli* DHFR, orthologous DHFRs do not cause any major perturbation in the intracellular purine and pyrimidine levels. In all experiments, sample size was three biological replicates. (**D**) Growth rate of *E. coli* over-expressing ADK, *E. coli* DHFR, and orthologous DHFRs 6, 11 and 16 in the presence of 0.05% arabinose is shown as a function of the intracellular dTTP levels. All data were normalized by growth rate of *E. coli* transformed with the empty plasmid. The data shows that fitness is tightly correlated to cellular metabolite levels (Pearson r = 0.99, p=0.0002).

The following source data and figure supplements are available for figure 4:

**Source data 1.** Raw and normalized data for metabolomics.

*Figure 4 continued on next page*

*Figure 4 continued*

**Source data 2.** Molecular properties of orthologous DHFR proteins.

**Figure supplement 1.** Activity ($v_{max}$) of over-expressed DHFR from whole cell lysate.

**Figure supplement 2.** The hexa-histidine tag on DHFR is neutral to fitness.

consumed in the last step of purine biosynthesis by PurH. Hence, partial inhibition of either of these paths can explain the metabolic imbalance and fitness cost associated with DHFR overexpression.

## DHFR overexpression toxicity is caused by PPI with essential enzymes.

De novo purine biosynthesis enzymes are essential for growth in minimal media, such as M9, but are non-essential in LB medium that contains purines (*Gerdes et al., 2003*; *Joyce et al., 2006*). This observation suggests that if inhibition of purine biosynthesis through PPI with overexpressed DHFR is partly responsible for GDT, DHFR overexpression would be less toxic in LB. Indeed, we found that

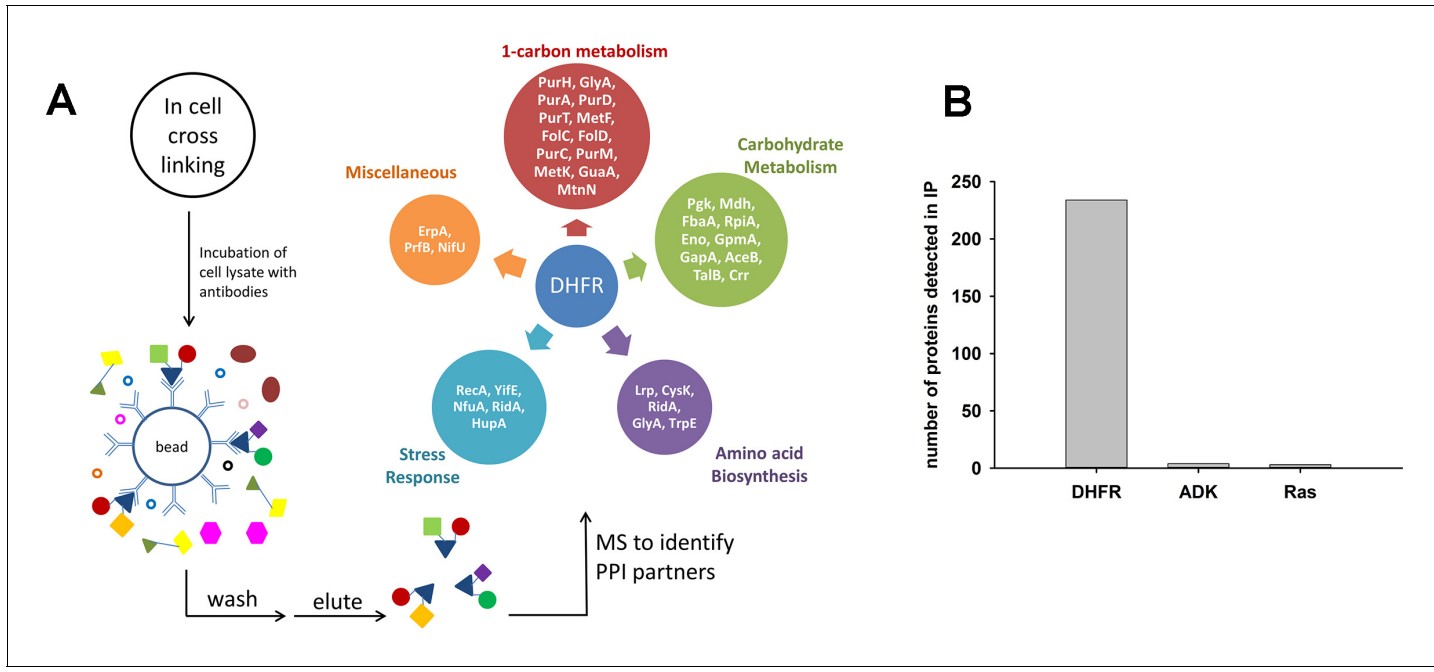

**Figure 5.** Detection of protein-protein interaction (PPI) partners of *E.coli* DHFR. (**A**) Schematics of the co-immunoprecipitation protocol. Proteins were cross-linked inside the cell using a cell-permeable cross-linker DSP (dithiobis(succinimidyl propionate)), and following lysis, the lysate was captured on anti-DHFR/anti-ADK/anti-Ras/anti-His-tag antibodies to fish out the protein of interest and its PPI partners. The interacting partners were subsequently identified using LC-MS/MS analysis. The observed interactome of overexpressed DHFR could be classified broadly into carbohydrate metabolism and 1-carbon metabolism (purine biosynthesis and amino acid metabolism groups). (**B**) Over-expressed *E. coli* DHFR picked out 234 proteins in total, while ADK and Ras detected only 4 and 3 proteins respectively. In case of DHFR, IP datasets were background subtracted using lysate from untransformed WT *E. coli* cells and DHFR3 (a MG1655 strain where the chromosomal DHFR has been replaced with DHFR from an orthologous bacteria (*Bershtein et al., 2015*) (also see SI Materials and methods for data analysis). In case of IP for ADK and Ras over-expression, background correction was done with lysate with only untransformed WT *E. coli* cells. Three biological repeat IP experiments with over-expressed *E. coli* DHFR and polyclonal anti-DHFR antibody (*Figure 5—source data 1*) had statistically significant correlation among themselves (average r = 0.8, p<0.001).

The following source data is available for figure 5:

**Source data 1.** Consolidated IP data for all proteins.

**Source data 2.** Consolidated list of binding partners of EcDHFR detected using anti-his and/or anti-EcDHFR antibodies.

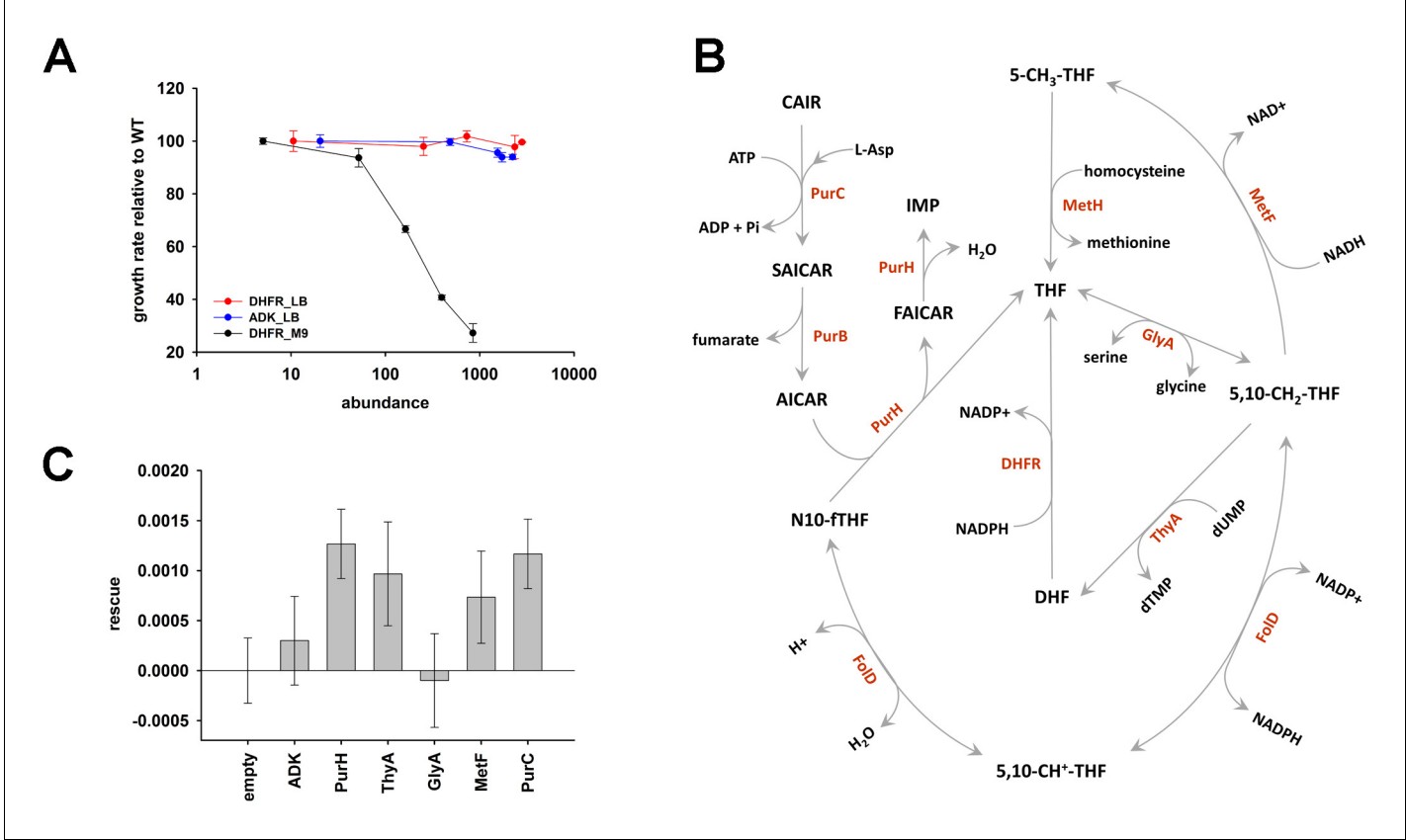

**Figure 6.** The role of purine biosynthesis and 1-carbon metabolism enzymes in DHFR over-expression toxicity. (A) Unlike supplemented M9 medium, over-expression of DHFR is not toxic in rich medium (LB) in terms of growth rate (Spearman r = −0.3, p=0.7), (note, however, that saturation ODs at 600 nm is lower in LB in comparison to growth in M9; see *Figure 6—figure supplement 1*). This indicates the role of conditionally essential genes like *purH, purC, metF, etc.* in determining the over-expression toxicity of DHFR. All data were normalized by growth rate of *E. coli* transformed with the empty plasmid. (B) Schematics of the 1-carbon pathway metabolic pathway. GlyA, ThyA, MetF are important enzymes that function immediately downstream of DHFR and utilize THF or its derivative. (C) Partial rescue of fitness of DHFR over-expressing *E. coli* by a dual-expression system. DHFR was expressed from a IPTG-inducible plasmid while PurH, ThyA, GlyA, MetF and PurC along with a negative control protein ADK were expressed separately from an arabinose-inducible pBAD system. Rescue factor is defined as $(\mu(DHFR+X) - \mu(DHFR+empty)) - (\mu(X) - \mu(empty))$, where $\mu$ is the growth rate, and X is the corresponding protein.

The following figure supplements are available for figure 6:

**Figure supplement 1.** OD vs time curves for DHFR over-expression in (A) rich media (LB) and (B) in M9 medium supplemented with amino acids.

**Figure supplement 2.** Effect of over-expression of purine biosynthesis and 1-carbon metabolism pathway proteins on growth rates of *E.coli* (A) alone and (B), (C) on the background of DHFR over-expression.

cells overexpressing DHFR do not show a reduction in 'growth rates' when grown in LB (*Figure 6A*), however there was a gradual drop in the saturation ODs at higher expression levels of DHFR (*Figure 6—figure supplement 1*). This might be due to DHFR interaction with other 1-carbon metabolism enzymes (GlyA, ThyA) or enzymes belonging to carbohydrate metabolism, namely phosphoglycerate kinase, fructose-biphosphate aldolase, which are essential even in LB (*Baba et al., 2006*). Further, we ran several complementation experiments in which we tested the effect of over-expression of enzymes involved in 1-carbon metabolism (schematically depicted in *Figure 6B*) on rescuing the DHFR overexpression toxicity. Most of these enzymes (with the exception of ThyA) were shown to interact with overexpressed DHFR in the IP/LC-MS/MS assay (*Figure 5*, *Figure 5— source data 2*). As shown in *Figure 6C*, overexpression of PurH, PurC, ThyA, and MetF resulted in a moderate yet significant alleviation of DHFR dosage toxicity, validating some of the interactions

detected in IP. However the rescue was only moderate, as most of these proteins themselves were toxic upon over-expression (*Figure 6—figure supplement 2*). A protein involved in tryptophan biosynthesis, TrpE, was detected in IP, which, if sequestered, might lead to a drop in tryptophan levels in the cell. Indeed addition of tryptophan to the growth medium partly rescued the growth defect of DHFR over-expression (*Figure 3B* and *Figure 3—figure supplement 1*), again validating the functional significance of the IP-detected interactions.

## *In vitro* analysis of DHFR interactions with its metabolic neighbors

Analysis of 1-carbon metabolism pathway suggested that effect of DHFR on several potential interaction partner proteins could explain, at least partially, the significant metabolic shifts and fitness effects of DHFR overexpression. Of them, we chose GlyA and PurH proteins for in depth *in vitro* analysis. GlyA works immediately downstream of DHFR in the folate pathway and uses THF to catalyze the reversible interconversion between glycine and serine (*Figure 6B*). Although complementation with overexpressed GlyA did not lead to partial rescue of fitness, presumably due to its intrinsic toxicity (*Figure 6—figure supplement 2*), addition of serine did lead to a partial alleviation of toxicity of overexpressed EcDHFR (*Figure 3—figure supplement 1*).

PurH is a bifunctional enzyme having IMP cyclohydrolase/aminoimidazole carboxamide ribonucleotide (AICAR) transformylase activities that catalyzes the last two steps in de novo purine synthesis (inosine-5′-phosphate biosynthesis path). A direct inhibition of both activities might explain the accumulation of AICAR and drop in IMP, AMP and GMP in the metabolite pools of overexpressing DHFR cells. Indeed, complementation with additional PurH protein showed a partial rescue of fitness (*Figure 6C*).

As a first step, we purified GlyA and PurH from *E. coli* and measured their interaction with *E. coli* and orthologous DHFRs *in vitro* using surface plasmon resonance (SPR). Both proteins were found to interact weakly with DHFR (*Figure 7—figure supplement 1* and *Figure 7—source data 1*). GlyA interacts with EcDHFR with a $K_D$ of 8 μM and with DHFR6 and 11 with $K_D$ of 5.3 μM and 4 μM respectively. PurH interacts with both EcDHFR and two orthologs (DHFR6 and DHFR11) with very similar $K_D$ in the low μM range (*Figure 7—figure supplement 1* and *Figure 7—source data 1*). ADK served as the negative control for all binding experiments and did not show any binding to GlyA and PurH. The similarity of the interaction between orthologous DHFRs and their interacting partners (GlyA and PurH) suggests that binding alone is insufficient to explain the observed selective toxicity of EcDHFR overexpression.

## *E. coli* DHFR affects activity of its interacting partners *in vitro*

The binding studies did not reveal significant differences in interaction of *E. coli* and orthologous DHFRs with GlyA and PurH. As a next step, we analyzed the *functional* effect of *E. coli* and orthologous DHFRs on the partner proteins. For GlyA, the formation of a ternary quinoid complex in the presence of pyridoxal phosphate (PLP), 5f-THF and glycine was used as a measure to determine its activity (*Schirch et al., 1985*) (see Materials and methods). While presence of ADK did not alter the ability of GlyA towards complex formation, *E. coli* as well as the orthologous DHFRs did show a concentration dependent inhibition of GlyA (*Figure 7A*). However, EcDHFR is clearly a more potent inhibitor of GlyA than its orthologs (*Figure 7B*). In addition, binding of EcDHFR to GlyA resulted in the largest increase in $K_D$ of 5f-THF binding (*Figure 7—figure supplement 2*).

We next assayed the AICAR transformylase activity of PurH in the presence of *E. coli* and the orthologous DHFRs, and the negative control protein ADK. EcDHFR was found to have the strongest concentration-dependent effect on both $k_{cat}$ and $K_M$ of AICAR transformylase activity of PurH (*Figure 7C,D* and *Figure 7—figure supplement 3*) with especially strong and pronounced detrimental effect on $K_M$. In contrast, addition of the same amount of DHFR11 did not affect the activity of AICAR transformylase to a significant degree. In summary, despite the similarity in binding parameters, the functional outcome of the interaction of DHFR with metabolically related enzymes GlyA and PurH is strikingly different for EcDHFR compared to the orthologs, and might, therefore, be a cause of selective toxicity of EcDHFR.

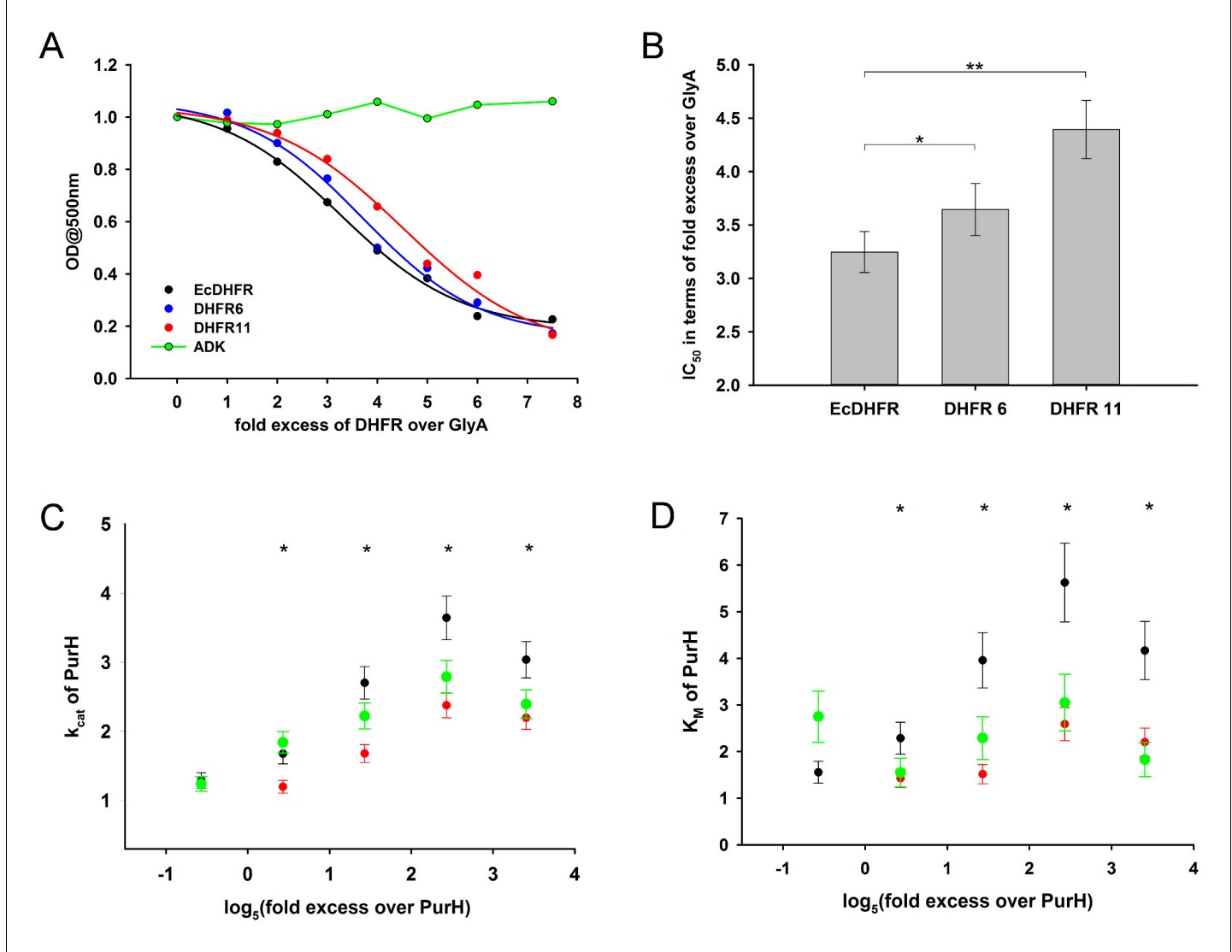

**Figure 7.** Compared to orthologous DHFRs, *E.coli* DHFR is a more potent inhibitor of GlyA and PurH. (**A**) Formation of a ternary complex (GlyA-PLP +glycine + 5f-THF) was monitored at 500 nm as a function of increasing amounts of different DHFRs and the negative control protein ADK. 20 μM GlyA was pre-incubated with varying concentrations of DHFRs and ADK (zero to 150 μM) before 0.2M glycine and 200 μM 5f-THF were added to it. All data were normalized by those in the presence of an equal volume of buffer. Data were fit to a 4-parameter sigmoidal function to extract the fold excess of DHFR required to achieve 50% inhibition ($IC_{50}$) as shown in panel (**B**). Though all DHFRs caused inhibition of GlyA compared to ADK, EcDHFR had a significantly lower $IC_{50}$ than DHFR6 (* denotes p-value<0.05) and DHFR11 (** denotes p-value<0.001), explaining its higher potency in sequestering GlyA and, hence, toxicity. Catalytic rate $k_{cat}$ (**C**) and Michaelis coefficient $K_M$ (**D**) of PurH for [10]Nf-THF measured in the presence of different DHFRs and ADK. All values were normalized relative to those of PurH measured in the absence of added protein. 250 nM of PurH was pre-incubated with varying concentrations of DHFRs or ADK (0.1 μM, 0.5 μM, 2.5 μM, 12.5 μM and 60 μM) and, subsequently, the initial rate of transfer of formyl group (AICAR transformylase activity) from [10]N-formyl THF to AICAR was measured at 298 nm. For determination of $k_{cat}$ and $K_M$ at each concentration of a protein, the concentration of [10]Nf-THF was varied from 20 μM to 1 mM, while AICAR concentration was fixed at saturation (500 μM). Each data point is an average of 3–5 independent measurements and the error bars represent standard deviation. For all proteins, $k_{cat}$ increased with increasing concentration of protein added, dropping off slightly at the highest concentration. The mechanism of this effect is not fully understood, but can be partially attributed to the generic crowding effect of proteins. However, only EcDHFR caused a concentration dependent increase in $K_M$ of PurH for [10]Nf-THF, thereby explaining its selective toxicity upon over-expression. Statistical analyses were done to compare $k_{cat}$ and $K_M$ values of EcDHFR (black) and DHFR11(red), and in all cases indicated by *, the p-value was <0.05.

The following source data and figure supplements are available for figure 7:

**Source data 1.** Kinetic parameters for binding of purified purH and glyA proteins to surface immobilized DHFRs by surface plasmon resonance (Biacore).

**Figure supplement 1.** Binding of purified PurH and GlyA proteins with EcDHFR, DHFR 6 and 11 in vitro detected using surface plasmon resonance.

*Figure 7 continued on next page*

*Figure 7 continued*

**Figure supplement 2.** Determination of equilibrium dissociation constant ($K_D$) of binding of 5f-THF to GlyA (**A**) Formation of a ternary complex (GlyA-PLP+glycine + 5f-THF) was monitored at 500 nm as a function of increasing amounts of 5f-THF in the presence of different DHFRs/ADK.

**Figure supplement 3.** Overlay of initial rate (maximum rate) of PurH activity as a function of $^{10}$Nf-THF concentration at different concentrations of added protein (**A**) EcDHFR (**B**) DHFR 11 and (**C**) ADK.

## The effect of PurH on DHFR activity at physiological concentrations is beneficial

The data so far suggest that over-expressed DHFR interacts with several proteins, including GlyA and PurH and potentially compromises their function. Importantly, the effect is the strongest when DHFR and the interacting partner are from *E. coli.* This observation points out to an intriguing possibility that the interactions might have a functional significance even at the basal expression levels of DHFR, but are too transient to be detected. To test this assumption, we analyzed the mutual effect of PurH and DHFR on each other's activity *in vitro* at physiological concentrations of both enzymes. While DHFR does not affect AICAR transformylase activity of PurH at physiological concentration (*Figure 7C,D*), we did observe a strong effect of PurH on activity of DHFR, which is highly specific to the *E. coli* variant (*Figure 8*). Both PurH and ADK resulted in a moderate increase in $k_{cat}$ of all DHFRs (*Figure 8A*), however the presence of PurH led to a significant decrease in $K_M$ of EcDHFR, while it had almost no or weak opposite effect on $K_M$ of the orthologues (*Figure 8B*). As a result, at physiological concentrations PurH significantly increases the catalytic efficiency ($k_{cat}/K_M$) of EcDHFR, whereas the activity of its orthologues remains unaffected (*Figure 8C*).

## Excess of DHFR converts transient interactions into permanent ones

Tetrahydrofolate, the product of DHFR activity is a precursor for 5,10-methylene THF, which is used for dTMP synthesis, as well as $^{10}$Nformyl-THF, which is eventually utilized by PurH in the last step of the de novo purine biosynthesis pathway to produce IMP. Hence, DHFR is functionally very closely connected to the family of purine/pyrimidine biosynthesis enzymes. Purine biosynthesis enzymes were found to be spatially localized forming the 'purinosome', in mammalian cells, but no evidence exists for *E. coli* (*An et al., 2008*; *French et al., 2016*; *Deng et al., 2012*). It is possible that at basal expression levels DHFR is a member of a dynamic 'metabolon' that facilitates channeling of tetrahydrofolate or its derivatives. However, when over-expressed, the toxicity is triggered by a stoichiometric imbalance of the complex which leads to conversion of these transient complexes to more permanent ones.

To assess the feasibility of such a scenario we use the Law of Mass Action to determine the fraction of DHFR and PurH in the monomeric form and in complexes:

$$[DHFR]^F + \frac{[DHFR]^F[PurH]^F}{K_D^{DHFR-PurH}} = [DHFR]^T$$
$$[PurH]^F + \frac{[DHFR]^F[PurH]^F}{K_D^{DHFR-PurH}} = [PurH]^T \qquad (1)$$

where square brackets denote the concentrations of enzymes, and superscripts *F* and *T* refer to free and total enzyme concentrations, respectively. $K_D^{DHFR-purH}$ is the equilibrium binding constant between PurH and DHFR. *Equation 1* represents simple stoichiometry relations for both enzymes.

To solve *Equation 1*, we used $K_D^{DHFR-purH} \approx 3\mu M$ determined in this study (*Figure 7—source data 1*), and known endogenous cellular concentrations of DHFR and PurH of $5 \times 10^{-8}$ and $2 \times 10^{-7}$ M respectively (corresponding to 50 and 200 molecules per cell [*Taniguchi et al., 2010*]). We found that at the endogenous abundance levels interaction between DHFR and PurH is transient with only 5% of DHFR found in complex with PurH and 1% of PurH is in complex with DHFR at any given time. However, when DHFR is ~1000 fold overexpressed, only 4% PurH remains free in cytoplasm, while the remaining 96% are complexed with DHFR. Similar estimates apply to GlyA.

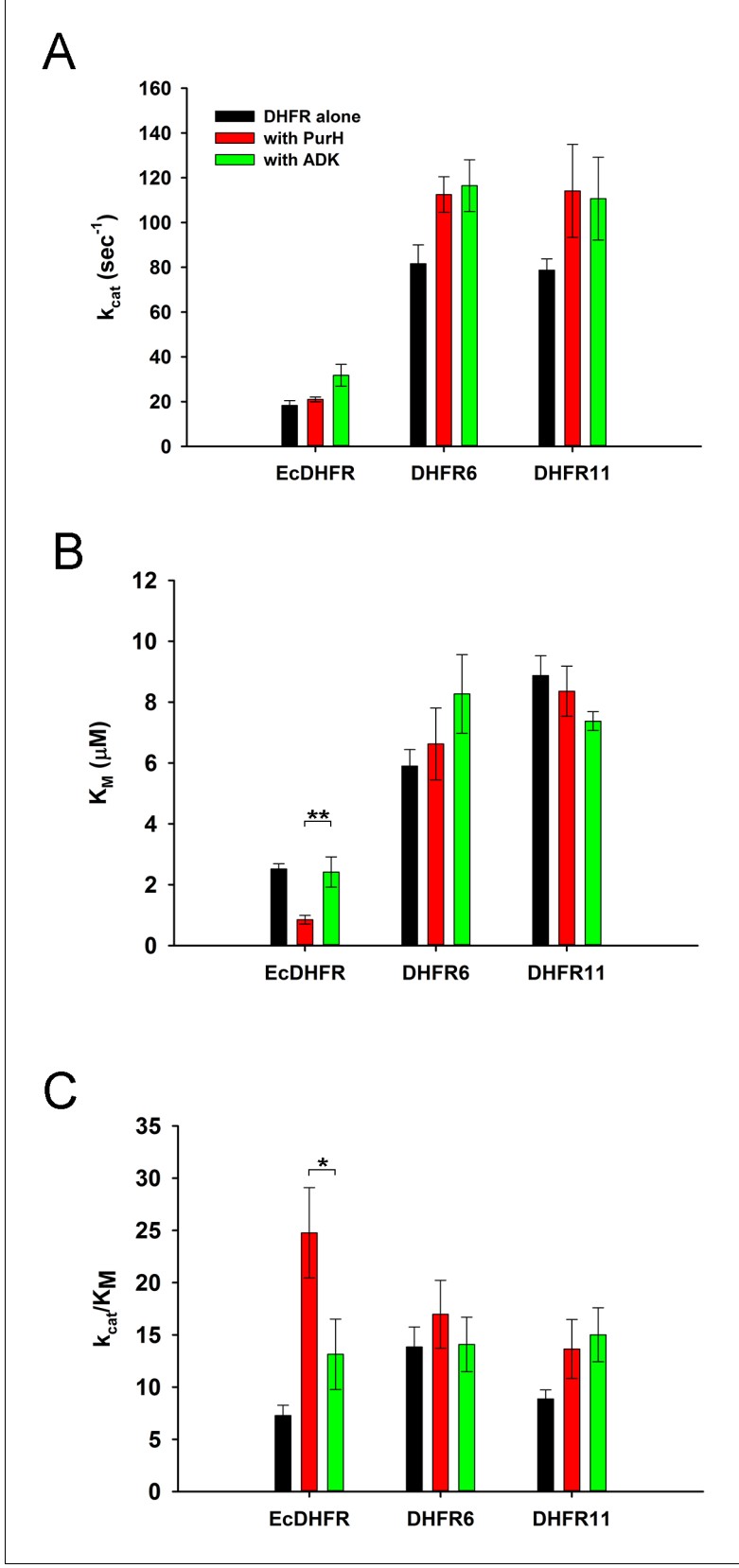

**Figure 8.** PurH has a selective beneficial effect on EcDHFR activity only (**A**) Catalytic rate ($k_{cat}$), Michaelis coefficient ($K_M$), and the catalytic efficiency ($k_{cat}/K_M$) of EcDHFR, DHFR6 and DHFR11 for dihydrofolic acid (DHF) in

*Figure 8 continued on next page*

*Figure 8 continued*

the presence of low concentrations of PurH.  10 nM DHFR were pre-incubated with 15 nM of PurH or ADK and, subsequently, the initial rate of conversion of NADPH to NADP+ was measured at 340 nm. For determination of $k_{cat}$ and $K_M$, the concentration of DHF was varied from 0.1 µM to 16 µM for EcDHFR and from 1 µM to 64 µM for DHFR6 and 11, while NADPH concentration was fixed at saturation (150 µM). Each data point is an average of 3–5 independent measurements and the error bars represent standard deviation. Both PurH and ADK resulted in an increase in $k_{cat}$ of all the DHFRs, however PurH caused a significant drop in $K_M$ and a concomitant increase in $k_{cat}/K_M$ for EcDHFR only. In all panels, * indicates p-value<0.05, while ** indicates p-value<0.001. Therefore, at physiological ratios of proteins, PurH is only beneficial for EcDHFR. This observation suggests an evolutionary functional relationship between PurH and EcDHFR at physiological concentrations. Orthologous DHFRs that have diverged during the course of evolution no longer have this benefit from *E. coli* PurH.

## Discussion

High-throughput studies showed that GDT is a common phenomenon affecting thousands of genes (*Sopko et al., 2006*; *Kitagawa et al., 2005*). However, despite its importance, there are no experimental 'case studies' that uncover molecular underpinnings of GDT. Here we carried out a study whereby we focus on a specific gene, *folA*, and delineate, in great detail and on multiple scales, the molecular mechanism of DHFR dosage toxicity in *E. coli*. We showed that fitness cost of DHFR overexpression could only be understood by integrating the metabolic and protein-protein interaction networks of the organism. Specifically, interactions between DHFR and several other metabolic enzymes, lead to a reduced flux through the purine, pyrimidine and amino acids synthesis paths. The imbalance in protein-protein interactions caused by overexpression of DHFR leads to a dead-end metabolic imbalance, such as the accumulation of AICAR – a precursor of inosine-5'-phosphate, and reduced growth. Overexpression of highly active but diverged orthologous DHFR caused only minor change in metabolite pools and fitness, indicating that neither protein burden nor metabolic cost per se can explain the endogenous DHFR dosage toxicity. The comprehensive IP/LC-MS/MS analysis confirmed by *in vitro* measurements of selected pairs of proteins clearly point out to spurious PPI formed between the overexpressed DHFR and several other metabolically and functionally related proteins in *E. coli* cytoplasm as the most probable cause of GDT.

The impact of overexpression on all levels of cell organization – molecular, systems, and phenotypic, dramatically differs between 'self', *i.e. E. coli*'s, and 'foreign' DHFR. This key finding clearly suggests the crucial role of evolutionary selection in shaping PPI at the whole proteome level. Remarkably, we found that at the physiological concentrations weak transient PPI can enhance the enzymatic activity of DHFR, provided that the interacting proteins are *from the same organism*. Further, we found that overexpressed 'foreign' DHFR pulls up many more interaction partners from *E. coli* proteome than does EcDHFR, suggesting that DHFR was subjected to selection against promiscuous mis-interactions in *E. coli* cytoplasm. Thus, evolution of the PPI network entails both positive *and* negative selection for transient interactions. Transient interactions between a select set of metabolically related proteins might be beneficial because they can enhance catalytic efficiency through ligand channeling. Conversely, they can be selected against to avoid massive non-functional promiscuous PPI. The weaker binding of EcDHFR to GlyA (*Figure 7—source data 1*) may also be indicative of a selection against transient interactions, as higher copy number enzymes, like GlyA, (expressed at ~10,000 copies/cell, [*Taniguchi et al., 2010*]) are more prone to mis-interact, comparatively to less abundant proteins, such as PurH (expressed at ~200 copies per cell [*Taniguchi et al., 2010*]). Earlier theoretical analyses envisioned both scenarios: selection against massive non-functional PPI (*Vavouri et al., 2009*; *Zhang et al., 2008*; *Heo et al., 2011*; *Levy et al., 2012*; *Yang et al., 2012*), and selection for interactions between metabolically related enzymes (*Huthmacher et al., 2008*; *Durek and Walther, 2008*; *Huthmacher et al., 2007*). This work provides experimental evidence that both these factors are indeed at play in evolution of PPI and metabolic networks.

It has long been established that the cell is not just a bag of enzymes, but there is a definite organization that enables enzymes to be compartmentalized. Seminal work carried out by Srere and colleagues first showed existence of such a functional supramolecular complex of sequential enzymes from Kreb's cycle in yeast, which led to coining of the term 'metabolon' (*Srere, 1987*, *1985*; *Robinson and Srere, 1985*). This was thought to be an efficient mechanism to channel common

intermediates through the pathway and thereby prevent them from escaping into solution. In other words, it would also increase the local concentration of substrate and help to carry out an efficient reaction with relatively low copy number of substrate molecules. During the same time, the term 'quinary structure' (the fifth level of protein structural organization) was coined by McConkey (*McConkey, 1982*) to describe transient protein-protein interactions. These interactions are not only low in thermodynamic stability ($K_D > 1$ µM), but also have a low kinetic barrier (*Wirth and Gruebele, 2013*), thereby making them amenable to modulation by cellular variations like metabolite concentrations, pH (*Cohen et al., 2015*; *Cohen and Pielak, 2016*), crowding, etc. Till date, the multi-enzyme complex metabolon remains to be the best example of a quinary structure. Recent studies have uncovered the role of quinary structures in modulating *in vivo* protein stability (*Monteith et al., 2015*). Our study unfolds a hitherto unexplored aspect of quinary structures: its role in gene-dosage toxicity.

Several authors hypothesized that sequestration of functional proteins into non-productive promiscuous PPI upon overexpression may cause GDT (*Vavouri et al., 2009*; *Heo et al., 2011*; *Yang et al., 2012*). Our study also shows that GDT of DHFR is triggered by potentially weak functional interactions gone 'awry' under the overexpression regime that causes a stoichiometric PPI imbalance and formation of permanent complexes in lieu of transient ones. However, unlike these studies, we show that the actual cause of the overexpression toxicity is the metabolic imbalance that ensues once the permanent PPIs are formed. Apparently, actual 'mis-interactions' are formed upon overexpression of orthologous DHFR proteins. In contrast to the previous studies, however, these interactions happen to be much less toxic than those picked by *E. coli* DHFR, at least at the experimental level of resolution. Nevertheless, weaker fitness effects of mis-interactions can affect evolution of the proteome to minimize promiscuous PPI (*Heo et al., 2011*; *Levy et al., 2012*), and our finding that non-endogenous DHFRs pulls out more interacting partners than EcDHFR points in this direction.

Broad distribution of protein abundances (*Ghaemmaghami et al., 2003*; *Ishihama et al., 2008*) and remarkable conservation of abundances of orthologous proteins between species (*Schrimpf et al., 2009*; *Laurent et al., 2010*) strongly suggests that protein abundance is a selectable trait. In contrast to protein stability, whose landscape is largely monotonic, at least for some proteins (*Rodrigues, 2016*; *Bershtein et al., 2012*; *Bloom et al., 2006*), fitness landscape for protein abundances is non-monotonic for many proteins, reaching an optimum at certain levels. While loss of fitness at low abundance is intuitively associated with loss of function, the GDT at high abundance is a system-level collective effect, related to interactions with multiple proteins as well as the link between PPI and metabolic networks, as shown here for DHFR. Apparently, specific functional demands dictate certain minimal amounts of proteins. Evolution of biophysical and biochemical parameters (activity, stability, and affinity to a functional protein and DNA partners) provide significant leverage to avoid prohibitively high expression levels which might cause GDT. Nevertheless, there is a certain evolutionary cost of maintaining high levels of activity and/or functional interaction strength due to supply of potentially deleterious mutations. Future studies will fully reveal the extent to which GDT affects evolved distributions of molecular properties of proteins and their abundances.

## Materials and methods

### Strains, media, and growth conditions

Strains used for down-regulation of DHFR were derivatives of *E. coli* MG1655 (CGSC #6300, ATCC #47076), while expression of pBAD plasmid in all cases were done in *E. coli* BW27783 cells (CGSC #12119) (*Khlebnikov et al., 2001*). Standard growth was conducted under the following conditions. Cells were grown from a single colony overnight at 37°C in M9 minimal salts supplemented with 0.2% glucose, 1 mM MgSO4, 0.1% casamino acids, and 0.5 µg/ml thiamine (supplemented M9 medium). Overnight cultures were diluted 1/100 and grown at 37°C. Growth rate measurements were conducted for 12 hr in Bioscreen C system (Growth Curves USA). OD data were collected at 600 nm at 15 min intervals. The resulting growth curves were fit to a bacterial growth model to obtain growth rate parameters (*Zwietering et al., 1990*). All experiments were done in triplicates and the error bars represent standard deviation. Growth rates values were typically associated with

2–3% standard deviation, and these errors did not change significantly upon increase of replicate number. Hence for convenience, a sample size of N = 3 was chosen for all growth rate measurements.

## Orthologous DHFRs

A BLAST analysis of *E. coli*'s DHFR amino acids sequence against mesophilic bacteria produced 290 unique DHFR sequences (*Bershtein et al., 2015*, *2013*). This dataset was used to select 35 DHFR sequences with amino acid identity to *E. coli*'s DHFR ranging from 29% to 96%. Out of these, in this study, we used DHFR sequences from five mesophilic bacteria *Listeria innocua* (DHFR 6), *Bordetella avium* (DHFR 11), *Leuconostoc mesenteroides* (DHFR 12), *Aeromonas hydrophila* (DHFR 13), *Clostridium cellulolyticum* (DHFR 16) and *Streptococcus dysgalactiae* (DHFR 17). These were cloned in an arabinose inducible pBAD plasmid between NdeI and XhoI sites, and each sequence contained a C-terminal hexa-Histidine tag to enable detection with anti-His antibody in Western blot. Non-His-tagged versions of DHFR6 and DHFR11 were also cloned and used for fitness measurements, to negate the possibility that presence of the histag can alter fitness parameters. For reference, *E. coli* DHFR was also cloned with and without the Histag. In this case also, presence of the tag did not change growth rate parameters.

## Gene dosage experiments

For over-expression, genes were cloned into pBAD plasmid under the control of arabinose inducible promoter, transformed into BW27783 cells, and subjected to varying arabinose concentrations (from 0% to 0.05%). The resulting intracellular abundance was calculated by Western Blot. EcDHFR, ADK and Ras proteins were detected using specific custom raised rabbit polyclonal antibodies (Pacific Immunology). Due to a profound sequence difference, anti-EcDHFR antibodies do not react with orthologous DHFR. To allow for the intracellular quantification of orthologous DHFR proteins, C-terminal His-tagged versions of orthologous and *E. coli* DHFR proteins were generated and detected using anti-His mouse monoclonal antibodies (Qiagen). All orthologous DHFRs were expressed from pBAD promoter using highest arabinose concentration of 0.05%. To validate that His-tag does not affect the observed fitness effects of endogenous and orthologous DHFRs, fitness effects of overexpression was measured with and without His-tags for all the DHFR proteins and found identical in all cases (*Figure 4—figure supplement 2*).

## Error bars for growth measurements

To assess significance of growth rates for orthologous DHFR over-expression as compared to EcDHFR (*Figure 3A and B*), the data for EcDHFR were fit to a 4-parameter sigmoidal function to obtain a continuous curve. Values of growth rate expected for the corresponding intracellular abundances of orthologs were obtained from this fitted line and were assigned a standard deviation of 3%. This expected growth rate was now compared to the experimentally observed growth rate for the orthologs.

Throughout the text, all p-values reported are derived from an unpaired t-test, unless otherwise stated.

## DHFR down-regulation

Generation of the strain with chromosomal controllable *folA* expression was previously described (*Bershtein et al., 2013*). Briefly, the operator sequence of the *lac* operon (*lacO*) was introduced upstream to −33 and −10 promoter signals of the chromosomal *folA* gene in a strain carrying Z1 cassette with *lacI* gene under constitutive promoter $P_{lacI}^q$ (*Lutz and Bujard, 1997*). Under the saturated IPTG concentration (0.6 mM), the resulted strain produced 20–25% of the basal DHFR expression level found in wild-type MG1655 strain, as determined by Western Blot. This abundance at 0.6 mM IPTG was scaled to 100%. Decrease in the IPTG concentration resulted in further drop of DHFR production.

## Intracellular protein abundance

Cells were grown in supplemented M9 medium for 4 hr at 37°C, chilled on ice for 30 min and lysed with 1×BugBuster (Novagen) and 25 units/ml of Benzonase. Intracellular DHFR, ADK and ras

amounts in the soluble fraction were determined by SDS-PAGE followed by Western Blot using rabbit anti-DHFR (*E. coli*)/anti-ADK/anti-ras polyclonal antibodies (custom raised by Pacific Immunology). For His-tagged orthologous DHFR, mouse anti-His antibody (Rockland Immunochemicals) was used for Western blots.

## Fluorescence microscopy

BW27783 cells transformed with pBAD-DHFR-GFP plasmid were grown overnight at 37°C from a single colony in supplemented M9 medium. Next day cells were diluted 1/100, and grown at 37°C for 4 hr. Cells were then pelleted and concentrated. 1 μl of a concentrated culture was mounted on a supplemented M9 + 1.5% low melting agarose (Calbiochem) pads and allowed to dry. Pads were then flipped on glass dish, and the images were acquired at room temperature with Zeiss Cell Observer microscope.

## Metabolite extraction

BW27783 cells transformed with pBAD-empty, pBAD-ADK or pBAD-DHFRs were grown in the presence of 100 μg/ml of Ampicillin and 0.05% arabinose till an OD of 0.5 (approx. 3.5 hr). The cells were then harvested by centrifugation at 4C, and washed three times with 1×M9 salts in water. Approximately 25 mg of cells were mixed with 300 μl of 80:20 ratio of methanol:water that had been pre-chilled on dry ice. The cell suspension was immediately frozen in liquid nitrogen followed by a brief thawing (for 30 s) in a water bath maintained at 25°C and centrifugation at 4°C at maximum speed for 10 min. The supernatant was collected and stored on dry ice. This process of extraction of metabolite was repeated two more times. The final 900 μl extract was spun down one more time and the supernatant was stored in −80C till used for mass spectrometry. For each construct, data reported are averaged over three independent colonies.

## LC-MS detection of metabolites

A Thermo q-Exactive Plus mass spectrometer coupled to a Thermo Ultimate 3000 HPLC was used to perform the LC-MS analysis of metabolites in biological samples and authentic chemical standards in negative ion mode. Electrospray source settings included a sheath gas flow rate was set of 35, auxiliary gas flow rate at 5 L/min, a capillary temperature of 250°C, and auxiliary gas temperature of 300°C. A calibration of the m/z range used was performed using the Thermo LC-MS Calibration mix immediately prior to the analysis. A scan range of 66.7–1000 m/z was used at a resolving power of 70,000 with alternating positive and negative ion mode scans. The chromatographic separation of metabolites was performed using hyrdophilic interaction liquid chromatography (HILIC) on a SeQuant ZIC-pHILIC column, 5 μm, polymer PEEK 150 mm x 2.1 mm column (EMD Millipore) at a flow rate of 0.1 mL/min. Mobile phase A was 20 mM ammonium bicarbonate with 0.1% ammonium hydroxide, and mobile phase B was acetonitrile. The mobile phase composition was started at 100% B, and subsequently decreased to 40% B over 20 min. The column was then washed at 0% B for five minutes before re-equilibration to 100% B over fifteen minutes. The extracted ion currents were plotted using a mass accuracy window of 5 ppm around the predicted monoisotopic m/z value of the molecular ion of each metabolite. The integrated area of each peak was used to determine the response for each metabolite at their specific retention time as determined by chemical standards.

## *In vivo* cross-linking, immunoprecipitation (IP) and detection by mass-spec

BW27783 cells transformed with pBAD-empty, pBAD-ADK or pBAD-DHFRs were grown in the presence of 100 μg/ml of Ampicillin and 0.05% arabinose for 4 hr. The cells were harvested by centrifugation at 4C, washed twice with 1×PBS and resuspended to a final OD of 5.0. For *in vivo* cross-linking, DSP (dithiobis(succinimidyl propionate)) was added to the cells to a final concentration of 2.5 mM (stock 250 mM in DMSO), and the reaction was allowed to proceed for 15 min at room temperature in a rotator/mixer. Extra cross-linker was quenched by adding 50 mM Tris, pH 8.0 and kept for additional 30 min at room temperature in the rotator. The cells were collected by centrifugation, the final weight was determined, and cells were stored at −20C.

For Co-IP, Invitrogen's Dynabeads Co-Immunoprecipitation Kit was used with some small modifications. For lysis, the frozen cell pellet was solubilized in 9:1 proportion of lysis buffer: weight of the

pellet (eg, 50 mg was resuspended in 450 μl buffer). The recommended 1×IP solution was supplemented with 1×Bugbuster, Benzonase (to a final concentration of 25 U/ml) and 100 mM NaCl and cell lysis was allowed to proceed for 20 min at room temperature in a rotator/mixer. The lysate was cleared by centrifugation at 4C for 30 min and applied directly to magnetic beads coated with the relevant antibody. The incubation time was 10 min at 4C. All washing steps followed were essentially as described in the kit. The bound proteins were eluted in 60 μl EB solution provided in the kit, neutralized with 1/10th volume of 1M Tris, pH 9.0, reduced with 10 mM TCEP and submitted for proteomics (protein identification by LC-MS/MS).

Binding partners for DHFR11 were detected using only anti-his antibodies, while those for EcDHFR were pulled out by both anti-his and polyclonal anti-DHFR antibodies in different experiments.

100 μg of affinity-purified polyclonal anti-DHFR, anti-ADK and anti-Ras antibodies were used per 10 mg of magnetic beads. For monoclonal anti-His antibody (Rockland Immunochemicals), 100 μg of affinity-purified antibodies were used.

## Analysis of IP data

For IP, two controls were used: MG1655 (DHFR expression from chromosome) and DHFR3. A modified *E. coli* MG1655 strain where the chromosomal DHFR has been replaced with DHFR from another mesophilic bacterium (*Pasteurella multocida*) is termed as 'DHFR3 strain' (**Bershtein et al., 2015**). The amino acid sequences of DHFR from *E. coli* and *Pasteurella multocida* have a low identity (52%), and hence DHFR3 protein does not bind the polyclonal anti-DHFR antibody. Using a paired t-test, we found that the average number of peptides for all observed proteins in the sample dataset (3 repeats of pBAD-EcDHFR) was significantly higher ($p<0.001$) than the average number of peptides for the same proteins in five control experiments (two repeats of MG1655 and three repeats of DHFR3). Hence for our analysis, we considered a particular protein to be a true positive interacting partner of DHFR if its average number of peptides in the sample was greater than the average number of its peptides in the control. As expected, the filtered dataset was also significantly different from the background data. All repeat experiments were biological replicates (different colonies of transformed cells).

## Rescue by protein complementation

To determine rescue by dual expression of DHFR and proteins belonging to the 1-carbon metabolism, a set of two compatible plasmids were used. pTRC plasmid contained DHFR under an IPTG-inducible promoter, had pBR322 origin of replication and conferred Ampicillin resistance. The other plasmid contained *purH/purC/metF/thyA/glyA* genes under an arabinose inducible pBAD-promoter, had pACYC origin of replication and conferred Chloramphenicol resistance. An empty pBAD plasmid served as the negative control. The overnight culture was diluted 1/100 in fresh medium containing $7.8 \times 10^{-4}$% arabinose (no IPTG was added) in addition to both antibiotics and growth rates were measured in Bioscreen C as described above.

## Binding studies by Biacore

Binding interaction studies were performed on a Biacore 3000 optical biosensor at 25°C. 700–900 resonance units of his-tagged *E. coli* DHFR or DHFR6 or DHFR11 protein were attached to the surface of a CM5 chip by standard amine coupling. A sensor surface on which 900 units of his-tagged ADK protein was immobilized served as a negative control for each binding interaction. Six different concentrations of histagged *E. coli* GlyA and PurH were flown over each channel. The running buffer was 25 mM potassium phosphate pH 7.4, 50 mM KCl, 0.2 mM EDTA and 1 mM β-mercaptoethanol, containing 0.005% P20 surfactant.

PurH concentrations used were 1.5 μM, 3 μM, 6 μM, 12 μM, 24 μM and 48 μM. Both binding and dissociation were measured at a flow rate of 30 μl/min. In all cases the sensor surface was regenerated between binding reactions by 1–2 washes with 5 mM NaOH at 30 μl/min. Each binding curve was corrected for nonspecific binding by subtraction of the signal obtained from the negative-control flow cell. The dissociation data was fit globally to obtain $k_d$ ($s^{-1}$). The RU obtained at the end of 100 s was plotted as a function of the concentration of analyte, and fitted to the following equation to derive the equilibrium dissociation constant ($K_D$).

$$RU = \frac{R_{max} \times conc}{conc + K_D}$$

### Activity measurement of purified PurH

For the AICAR transformylase activity, $^{10}$Nformyl-THF was prepared from (6R,S)−5-formyl-THF (Schircks Laboratories, Jona, Switzerland) essentially as described in (*Xu et al., 2004*). For the assay, 250 nM of PurH was incubated alone or with 0.1, 0.5, 2.5, 12.5 and 60 µM of DHFRs or ADK for 30 min at room temperature in buffer (50 mM Tis-HCl pH 7.4, 50 mM β-mercaptoethanol and 25 mM KCl) containing varying amounts of $^{10}$Nf-THF in a total volume of 95 ul. The reaction was initiated by adding 5 µl of 10 mM AICAR (final concentration 0.5 mM) and the increase in absorbance at 298 nm was monitored.

### Activity assay for GlyA

20 µM of GlyA in buffer (50 mM Tis-HCl pH 7.4, 50 mM β-mercaptoethanol and 25 mM KCl) was pre-incubated with varying concentrations of DHFRs/ADK (zero to 150 µM) before addition of 0.2M glycine and 200 µM 5f-THF (*Schirch et al., 1985*) and the amount of the ternary complex formed was monitored at 500 nm. To determine of $K_D$ for 5f-THF in the presence of added DHFR/ADK, the concentration of the latter was fixed at 60 µM. The procedure followed is essentially described in (*Fu et al., 2003*).

### Activity assay for DHFRs

10 nM DHFR in MTEN buffer (50 mM 2-(N-morpholino) ethanesulfonic acid, 25 mM tris (hydroxymethyl) aminomethane, 25 mM ethanola- mine, and 100 mM sodium chloride, pH 7) were pre-incubated with 15 nM of PurH or ADK and, subsequently, the initial rate of conversion of NADPH to NADP+ was measured by monitoring absorbance at 340 nm. The concentration of dihy- drofolate (DHF) was varied from 0.1 µM to 16 µM for EcDHFR and from 1 µM to 64 µM for DHFR6 and 11, while NADPH concentration was fixed at saturation (150 µM).

## Acknowledgements

We thank Bharat Adkar for valuable discussions and help with analysis of the proteomics and metab- olomics data, Adrian Serohijos and Michael Manhart for valuable discussions and suggestions. This work was funded by NIH RO1 GM111955.

## Additional information

### Funding

| Funder | Grant reference number | Author |
| --- | --- | --- |
| National Institute of General Medical Sciences | GM111955 | Eugene I Shakhnovich |

The funders had no role in study design, data collection and interpretation, or the decision to submit the work for publication.

### Author contributions

SBh, SBe, Conception and design, Acquisition of data, Analysis and interpretation of data, Drafting or revising the article; JY, TA, SAT, Acquisition of data, Analysis and interpretation of data; AIG, Analysis and interpretation of data, Contributed unpublished essential data or reagents; EIS, Con- ception and design, Analysis and interpretation of data, Drafting or revising the article

### Author ORCIDs

Sanchari Bhattacharyya, http://orcid.org/0000-0003-3421-4755
Amy I Gilson, http://orcid.org/0000-0003-2046-3603
Eugene I Shakhnovich, http://orcid.org/0000-0002-4769-2265

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
