## [Decision Letter]

Thank you for submitting your article "Transient protein-protein interactions perturb *E. coli* metabolome and cause gene dosage toxicity" for consideration by *eLife*. Your article has been reviewed by three peer reviewers, and the evaluation has been overseen by a Reviewing Editor and Naama Barkai as the Senior Editor. The reviewers have opted to remain anonymous.

The reviewers have discussed the reviews with one another and the Reviewing Editor has drafted this decision to help you prepare a revised submission.

Summary:

The manuscript of Battacharya et al. studied the cause of gene dosage toxicity, i.e. the toxic effect of overexpression of a its protein product. The study uses the enzyme DHFR in *E. coli* as a model system and shows that:

– there is a specific dosage toxicity effect for that protein absent for some other proteins overexpressed at similar levels

– this effect is accompanied by a change in metabolite levels (metabolic imbalance)

– toxicity is related to weak binding of DHFR to essential enzymes.

Throughout the study, the authors compare DHFR proteins from different bacteria and claim that the effect is specific to the *E. coli* enzyme, indicating that the weak interactions underlying the toxicity effect is evolvable and likely tuned to the a desired level.

The problem discussed here is in important one and the results presented are of general interest. However, the reviewers also noticed some points that need improvement. Specifically, the claim that the effect is specific to the *E. coli* enzyme is not convincing and thus the evolutionary interpretation is not very strongly supported.

Essential revisions:

Main issue: The comparison between the enzymes from *E. coli* and from other bacteria is a central aspect of the paper. However, the heterologous enzymes contain his tags, while the native one does not. It is thus possible that the specificity to the *E. coli* enzyme is due to this technical difference, which would invalidate the comparison of the proteins and the evolvability claims. Without proper controls that deal with the possible influence of the his tag, this paper could not be reconsidered for publication.

Additional issues raised by the reviewers:

1) Overexpression vs. depletion (Figure 1): It should be mentioned that different strains are used. Do they have the same WT growth rate? Does the empty plasmid affect growth?

2) Overall, the description of the experimental methods needs to be improved and more detail needs to be given. In addition to the two points above:

- Figure 2: No methods description is provided for the fluorescence microscopy, nor is the DHFR overexpression level mentioned in the legend (or elsewhere).

- The full metabolomics dataset is not included.

- In [Supplementary-material SD4-data] (Consolidated IP data for all proteins), what is represented by the column titled "BW"?

3) The discussion of relevant previous work (e.g. by the late Paul Srere) as well as current work on quinary structure needs to be improved.

4) A discussion of why no growth defect was observed for adenylate kinase overexpression would also be helpful. Can the difference between the proteins be attributed to their known properties? Can some properties be ruled out as cause of the difference? And does the difference indicate that toxicity due to weak interactions is rare?

[Editors' note: further revisions were requested prior to acceptance, as described below.]

Thank you for resubmitting your work entitled "Transient protein-protein interactions perturb *E. coli* metabolome and cause gene dosage toxicity" for further consideration at *eLife*. Your revised article has been favorably evaluated by Naama Barkai (Senior editor), a Reviewing editor, and one reviewer.

The manuscript has been improved but there are some remaining issues that need to be addressed before acceptance, as outlined below:

The main revision requested was to address possible artefacts due to the use of his tags in the heterologous enzymes. The authors have added crucial control data comparing the growth effect of different DHFR proteins with and without a his tag. Presence or absence of the tag does not show a difference, while a big difference is seen between the *E. coli* enzyme (with or without his tag) and the heterologous enzymes with or without enzymes. This rules out that the difference between the native and heterologous enzymes is cause by the difference in tags.

The only small, but important change that is strongly suggested here is to explicitly state the level of induction of the pBAD promoter (probably at maximal expression, but this is not stated and for the heterologous enzymes it cannot be inferred from the data). As this control is crucial, being as clear as possible about the conditions is highly recommended.

The other questions of the reviewers have been clarified in the revision.

---

## [Author Response]

*Essential revisions:*

*Main issue: The comparison between the enzymes from E. coli and from other bacteria is a central aspect of the paper. However, the heterologous enzymes contain his tags, while the native one does not. It is thus possible that the specificity to the E. coli enzyme is due to this technical difference, which would invalidate the comparison of the proteins and the evolvability claims. Without proper controls that deal with the possible influence of the his tag, this paper could not be reconsidered for publication.*

We thank the reviewer(s) for highlighting this important issue and agree that it was not clarified explicitly in the manuscript. Since the orthologous DHFRs have very low sequence identity with *E. coli* DHFR, the polyclonal antibodies raised against *E. coli* DHFR do not recognize the orthologous proteins To be able to perform the intracellular quantifications of DHFR abundances, to conduct the IP experiments, and to have an efficient protein purification tool for the in vitro measurements, DHFR proteins were His-tagged at their C-terminus. To validate that the presence or absence of the His-tag in either *E. coli* or orthologous DHFR proteins do not affect the observed fitness effects of DHFR overproduction, we generated both tagged and untagged versions of EcDHFR, and orthologous DHFR6 and DHFR11, overexpressed these proteins, and measured the resulting growth rates. We found that the presence or absence of His-tag had no effect on *E. coli* DHFR gene dosage toxicity – both protein versions showed identical levels of toxicity. Similarly, presence or absence of His-tag did not affect at all the lack of toxicity of orthologous DHFR proteins. These data are now included in the new Figure 4—figure supplement 2. Additionally, we now emphasize this point in the Methods section, and in captions to Figure 1 and Figure 4.

*Additional issues raised by the reviewers:*

*1) Overexpression vs. depletion (Figure 1): It should be mentioned that different strains are used. Do they have the same WT growth rate? Does the empty plasmid affect growth?*

For over-expression experiments, *E. coli* strain BW27783 was transformed with arabinose inducible pBAD plasmid and all experiments were done at 37°C. For the downregulation experiments, *E. coli* MG1655 strain was engineered to contain the lac operator upstream of the -10 and -33 promoter signals of the chromosomal *folA* gene (termed B3 strain) and growth rate and intracellular abundance measurements were done at 30°C. Since experiments were done at different temperatures for over-expression and down-regulation, the absolute values of growth rates are not comparable. However, it is important that at 30°C, growth rates of B3 strain (under saturating IPTG concentration of 0.6mM) and the parent MG1655 strain were indistinguishable. These data are now shown as a consolidated list of growth rates in [Supplementary-material SD1-data].

However, we would like to emphasize that the temperature of the experimental study or the absolute magnitude of growth rates of the strains are not important. Rather it is the dependence of growth rate on intracellular protein abundance and its dynamics (e.g., bell like shape curve, and different slopes for up and down regulations) that are most important messages that we wish to convey from Figure 1.

The empty plasmid is neutral and does not affect growth of BW27783. This data is already part of Figure 1—figure supplement 1.

*2) Overall, the description of the experimental methods needs to be improved and more detail needs to be given. In addition to the two points above:*

*- Figure 2: No methods description is provided for the fluorescence microscopy, nor is the DHFR overexpression level mentioned in the legend (or elsewhere).*

*- The full metabolomics dataset is not included.*

*- In [Supplementary-material SD4-data] (Consolidated IP data for all proteins), what is represented by the column titled "BW"?*

We thank the reviewer for pointing this out. A description of fluorescence microscopy is now given in the Materials and methods section. The DHFR over-expression level for these experiments was ~350-fold and this is now included in the legend to Figure 2 as well.

The full metabolomics dataset is now included as [Supplementary-material SD2-data]

BW represents untransformed BW27783 cells which do not have any plasmid. We understand this is confusing, and hence now this column is renamed as “BW27783-no plasmid”.

*3) The discussion of relevant previous work (e.g. by the late Paul Srere) as well as current work on quinary structure needs to be improved.*

Our discussion was mainly focused on the evolutionary implications of weak protein-protein interactions, and indeed we missed out on the relevant biochemical aspects. We thank the reviewer for this important pointer. A detailed discussion of relevant literature on metabolon and quinary complexes is now included in the Discussion section of the manuscript.

*4) A discussion of why no growth defect was observed for adenylate kinase overexpression would also be helpful. Can the difference between the proteins be attributed to their known properties? Can some properties be ruled out as cause of the difference? And does the difference indicate that toxicity due to weak interactions is rare?*

Our IP experiments show that Adenylate Kinase (Adk) does not pull out any significant number of proteins. In other words, over-expressed Adk does not interact with other proteins in the cell, and we believe this must the reason for the lack of growth defect. However which specific properties of the protein would decide its propensity to interact with other proteins is immediately unclear from the present study. A possible reason might be the following: our study shows that DHFR engages in weak interactions with its functional neighbors, possibly through formation of a metabolon. This is especially important for its function since DHFR is part of a pathway of 1-carbon metabolism, hence formation of a metabolon would facilitate exchange of ligands. In contrast, Adk is a stand-alone enzyme that is fully reversible, and thereby maintains the balance of adenylate currencies (ATP, ADP, AMP) in the cell. Hence it is possible that Adk does not have a significant number of functional neighbors. To test if this is universally true for all stand-alone enzymes can be a topic of an interesting future study.

[Editors' note: further revisions were requested prior to acceptance, as described below.]

*The main revision requested was to address possible artefacts due to the use of his tags in the heterologous enzymes. The authors have added crucial control data comparing the growth effect of different DHFR proteins with and without a his tag. Presence or absence of the tag does not show a difference, while a big difference is seen between the E. coli enzyme (with or without his tag) and the heterologous enzymes with or without enzymes. This rules out that the difference between the native and heterologous enzymes is cause by the difference in tags.*

*The only small, but important change that is strongly suggested here is to explicitly state the level of induction of the pBAD promoter (probably at maximal expression, but this is not stated and for the heterologous enzymes it cannot be inferred from the data). As this control is crucial, being as clear as possible about the conditions is highly recommended.*

In response to your request we have added statements indicating that all data for comparison between his-tagged and non-his tagged versions of native and heterologous DHFR has been obtained at highest level of induction of the pBAD promoter used in our study (0.05% of arabinose). The following sections of the manuscript have been updated to reflect this information:

1. Materials and methods section

2. Legend to Figure 4

3. Legend to Figure 4—figure supplement 2

4. [Supplementary-material SD3-data] now indicates the abundances of *E. coli* DHFR and orthologous DHFRs at the highest inducer concentration of 0.05% arabinose.